# Neuroprotective Effect of Angiopoietin2 Is Associated with Angiogenesis in Mouse Brain Following Ischemic Stroke

**DOI:** 10.3390/brainsci12111428

**Published:** 2022-10-24

**Authors:** Ling-Ling Lv, Yi-Ting Du, Xiao Chen, Yu Lei, Feng-Yan Sun

**Affiliations:** 1Department of Neurobiology and State Key Laboratory of Medical Neurobiology, School of Basic Medical Sciences, Shanghai Medical College, Fudan University, Shanghai 200032, China; 2Institute for Basic Research on Aging and Medicine of School of Basic Medical Sciences and National Clinical Research Center for Aging and Medicine, Huashan Hospital, Hanghai Medical College, Fudan University, Shanghai 200032, China

**Keywords:** ischemic stroke, neuroprotective, Angiopoietin 2, Angiogenesis, CD34

## Abstract

Angiogenic factors play an important role in protecting, repairing, and reconstructing vessels after ischemic stroke. In the brains of transient focal cerebral ischemic mice, we observed a reduction in infarct volume after the administration of Angiopoietin 2 (Angpt2), but whether this process is promoted by Angpt2-induced angiogenesis has not been fully elaborated. Therefore, this study explored the angiogenic activities, in reference to CD34 which is a marker of activated ECs and blood vessels, of cultured ECs in vitro and in ischemic damaged cerebral area in mice following Angpt2 administration. Our results demonstrate that Angpt2 administration (100 ng/mL) is neuroprotective by significantly increasing the CD34 expression in in vitro-cultured ECs, reducing the infarct volume and mitigating neuronal loss, as well as enhancing CD34^+^ vascular length and area. In conclusion, these results indicate that Angpt2 promotes repair and attenuates ischemic injury, and that the mechanism of this is closely associated with angiogenesis in the brain after stroke.

## 1. Introduction

The early acute phase of stroke encompasses multiple pathophysiological changes, such as elevated oxidative stress and apoptosis, impeded energy supply and glutamate reuptake, interrupted ionic gradients and calcium homeostasis, etc [1]. The neurovascular unit (NVU), consisting of endothelial cells (ECs), vascular smooth muscle, astroglia and microglia, neurons and associated tissue matrix proteins [2], is conducive to the brain’s vasculature, improving the outcome after strokes. ECs combined with angiogenic factors are particularly instrumental in this process.

Besides sensing and responding to blood flow, in addition to being a constitutive part of the blood–brain barrier (BBB), ECs also contribute to repair in hypoxia-ischemic neonatal mice by promoting excitatory synaptogenesis [3]. Moreover, angiogenic factors also play important roles after stroke. For example, vascular endothelial growth factors significantly promote angiogenesis and reduce infarct volume in adult rat brains after cerebral ischemia [4]. The pigment epithelium-derived factor, another angiogenic factor, modulates BBB permeability and attenuates lesion volume expansion at an acute phase of ischemia [5]; intriguingly, Angiopoietin 2 (Angpt2) has been observed to have a similar effect [6].

Angiogenic factor Angpt2, one of the principal ligands of Tie2 receptor and secreted by endothelial cell exosomes, has a higher expression level in endothelial tip cells than in endothelial stalk cells [7,8]. Tip cells are the leading cells of the sprouts, which direct the sprouting process in angiogenesis. Angpt2 regulates angiogenesis through Tie2 and integrin signaling [9], and also activates integrins on the cell surface, causing the phosphorylation of integrin-aligning proteins. This in turn activates RAC1, promoting cell migration and sprouting angiogenesis [10]. Moreover, high concentrations of Angpt2 can enhance endothelial survival rate through the PI3K/AKT pathway [11]. Notwithstanding Angpt2′s angiogenesis regulating ability, whether Angpt2 could ameliorate cerebral ischemic damage by enhancing angiogenesis, and its effectiveness, still need to be elaborated.

CD34, a transmembrane phosphoglycoprotein, was first identified in 1984 on hematopoietic stems and progenitor cells [12]. Not only are circulation CD34^+^ cells recognized as endothelial progenitors, effective in treatment of cardiovascular diseases, but local CD34^+^ cells are also believed to be crucial for angiogenesis [13].

Therefore, the aim of this study was to evaluate whether the protective effect of Angpt2 against ischemic stroke correlates with enhanced angiogenesis by utilizing the length and area of CD34^+^ blood vessels as the deciding references for angiogenic activities after cerebral ischemia [12,14,15].

In summary, we used the transient middle cerebral artery occlusion (MCAO) model of cerebral ischemic injury in mice combined with Angpt2 administration to investigate the function of Angpt2 in stroke. Our results showed that the administration of Angpt2 reduced the infarct volume and neuronal loss, the mechanism of which was associated with magnified CD34^+^ vascular length and area after stroke.

## 2. Materials and Methods

### 2.1. Animals

8–12-week-old mature male C57BL/6 mice (weighting 25–30 g) were obtained from the Shanghai Experimental Animal Center of the Chinese Academy of Sciences. All animals care protocols and experimental procedures were approved by the Institutional Animal Care Committee of Fudan University Shanghai Medical College Committee (Permit Number: 20180302-108). All mice were housed at the Animal Facility of Fudan University under a 12 h light/dark cycle and had access to food and water. All efforts were made to minimize animal suffering and reduce the number of animals used. Animals were randomly assigned to the experimental groups.

### 2.2. Middle Cerebral Artery Occlusion (MCAO) and Laser-Doppler Blood Flow Analysis

Focal cerebral ischemia was induced based on previous reports [16]. In brief, a 6-0 nylon monofilament suture with a rounded tip (Jialing Biotech CO, LTD, Guangzhou, China) was introduced into the left external carotid artery lumen and advanced into the internal carotid artery to occlude the middle cerebral artery blood flow for 35 min. In each animal (thermoregulated and anesthetized), the calvarium was exposed by incision, and the blood flow of the left middle cerebral artery was measured using a Laser Doppler blood flow analyzer (Periflux system 5000, Perimed, Stockholm, Sweden). The blood flow was recorded during the operation and then normalized to the baseline in each mouse.

### 2.3. TTC Staining

TTC staining was performed to show infarct area based on previous reports [17]. Briefly, three days after reperfusion, the animals were anesthetized prior to their brains being carefully dissected. The brains were washed in previously cooled phosphate-buffered saline and serial coronal sections (1 mm thick) were made from the olfactory bulb to the cerebellum. The sections were placed in 2% 2,3,5-triphenyltetrazolium chloride (TTC, T8877, Sigma-Aldrich, Darmstadt, Germany), dissolved in phosphate buffered saline and stained for 10 min at 37 °C in the dark. The sections were fixed with 4% paraformaldehyde (A500684-0500, Sangon Biotech, Shanghai, China) for 5 min at room temperature and then scanned for further analysis.

### 2.4. Neurological Deficit Measurements and Scores

One and three days after MCAO, neurological functions of the BSA-MCAO and Angpt2-MCAO groups were observed and evaluated according to the method of the Zea Longa test [18], using a five-point scoring system as previously described [19]. The score derived from this is in the ascending order, indicating neurological deficit severity from the normal to the most severe: 0 = no observable neurological deficit when mouse is suspended by its tail; 1 = showing slight deficit, with flexion of torso and contralateral forelimb when mouse is suspended by the tail; 2 = moderate deficit, with mouse circling to the contralateral side when its tail held on the flat surface; 3 = severe deficit, with mouse leaning to the contralateral side at rest; 4 = most severe deficit, with mouse showing no spontaneous motor activity and depressed levels of consciousness. Neurological assessment was made by an observer blind to the treatment conditions.

### 2.5. Brain Micro-Endothelial Cells (BMECs) Preparation

The BMECs were prepared from C57BL/6 mice based on previous reports [3,20]. After 80% confluence, secondary passage-cultured cells had Angpt2 (100 ng/mL) added. Then, the cells were harvested for CD34 expression analysis after different incubation times (30 min, 2 h, 6 h).

### 2.6. Western Blot

As described previously [21], the protein of BMECs and striatum of mouse brains was homogenized using lysis buffer (P0013J, Beyotime, Shanghai, China) with a protease inhibitor (04693159001, 04906845001, Roche, Rotkreuz, Switzerland). The sample supernatants were obtained, denatured and separated on a 10% sodium dodecyl sulfate polyacrylamide gel. Then, they were transferred to a polyvinylidene difluoride membrane (BIO-RAD). Membranes were blocked with 10% non-fat milk in TBST (10 mmol/L Tris-HCl, 150 mmol/L NaCl, 1% Tween-20, pH7.4) for 2 h at room temperature and incubated with rabbit anti-CD34 antibodies and mouse anti-β-actin. After washing with TBST, the membranes were incubated with antirabbit IgG-HRP and anti-mouse IgG-HRP for 1 h at room temperature. The membranes were visualized using the Western Lightning Plus-ECL kit (NEL105001EA, PerkinElmer, Akron, OH, USA), then scanned and visualized by ChemiDoc Touch Imaging System, version 5.2.1 (BIO-RAD, Hercules, CA, USA). Relative intensities were normalized to the intensity of corresponding bands for β-actin within a single gel. A band of samples from contralateral striatum was taken to be equal to 1, and ipsilateral striatum band was then quantified relative to the contralateral striatum.

### 2.7. Angpt2 Injection

To evaluate the effect of Angpt2 on CD34 expression after stroke, the BSA or Angpt2 (100 ng/μL, 623-AN, RD systems) in 2 μL PBS were stereotaxically injected into the left striatum, as previously described, with slight modifications (coordinates from Bregma: AP 0.5 mm, ML 2.0 mm, DV 2.8 mm from the skull surface) [3]. Mice were subjected to MCAO operation at 12 h after BSA or Angpt2 injection and sacrificed 3 days after operation.

### 2.8. Brain Sections Preparation

Mice were deeply anesthetized and cardially perfused with 0.9% sodium chloride and 4% paraformaldehyde, as previously described [16,22]. The brains were removed and post-fixed in 4% paraformaldehyde for 12 h and then immersed sequentially in 20% and 30% sucrose solutions in 0.1 mol/L phosphate buffer (pH 7.4) until they sank. The coronary brain sections at the thickness of 30 μm were stored at −20 °C in cryoprotectant solution.

### 2.9. Cresyl Violet Staining

For cresyl violet staining, coronal sections (30 μm) separated by 330 μm were mounted on slides. Then, the slides were immersed in 75, 85, 95, 95, 100, 100% ethanol for 8 min each and were immersed back through the 95, 85, 75, 50% ethanol and double-distilled water for 8 min in each. Next, they were placed in a cresyl violet solution and then dehydrated again in 75, 85, 95, 100% ethanol for 1 min each. The slides were placed in 50% xylene (in 100% ethanol) and 100% xylene for another 30 min, and then coverslipped. Infarct volume was evaluated as previously described [19,23], quantified using the Image-J 1.37c software (National Institutes of Health, Bethesda, MD, USA), and then estimated using following formula: corrected infarct volume = [left hemisphere area—(right hemisphere area—right hemisphere infarct area)] * (thickness of section + distance between sections). To give a three-dimensional approximation of the total infarct volume, corrected infarct volumes of individual brain sections were shown through distance from the bregma [19,24].

### 2.10. Immunofluorescence and Immunohistochemistry Staining

Sections of each mouse brain between 0.5 mm and 0 mm from the bregma were prepared and immunostained, as previously described [4,25,26]. To observe CD34 expression in mouse brains after MCAO, brain sections were co-incubated with rabbit anti-CD34 antibody (CY5196, Abways Technology, Shanghai, China) overnight at 4 ℃. After washing with 0.01 mol/L PBS, immunoreactive signals were correspondingly revealed by donkey antirabbit IgG-Alexa Fluor 488 (A21206, Thermo Fisher Scientific, Waltham, MA, USA) secondary antibodies. Fluorescent signals in the brain sections were detected using the TCS SP8 confocal laser scanning microscope (Leica, Heidelberg, Germany) at excitation and emission wavelengths of 488 nm and 525 nm (Alexa Fluor 488). The length and area of vessels were evaluated in the striatum using Image-J 1.37c software (National Institutes of Health). Three brain sections of each mouse were selected, and 3 areas close to the injection point in each section were measured and analyzed [3]. For detecting CD34 and neuron distribution in whole brains after MCAO, brain sections were coincubate with rabbit anti-CD34 antibody or rabbit anti-NeuN antibody overnight at 4 °C, followed by incubation with antirabbit biotinylated secondary antibody (Vector Laboratories, CA, USA) and the avidin–biotin–peroxidase complex (Vector Laboratories, Burlingame, CA) for 1 h at 37 °C. Immunoreactivity was detected with 0.05% diaminobenzidine in a Tris-HCl buffer (0.1 M, pH 7.6) containing 0.03% H_2_O_2_.

### 2.11. Statistical Analysis

All data were presented as the means ± SEM and were compared either by one-way ANOVA with Tukey’s post hoc test for multiple groups or by a two-sample ++ test between two groups. Statistical analyses were determined using Origin, version 9.0 (OriginLab Corp, Northampton, United States). Statistical differences were defined as *p* < 0.05.

## 3. Results

### 3.1. Angpt2 Exerted a Neuroprotective Effect by Reducing the Brain Infarction in Mice after Ischemic Stroke

To confirm whether Angpt2 administration could improve damage caused by ischemic injury, we first measured the blood flow before and during MCAO, analyzing brain sections through TTC and CV staining to assess the validity of the stroke model. Using Laser Doppler flowmetry, we observed a drop of over 80% in blood flow compared to the baseline (100%) at the time of embolus insertion. The decreased flow continued and was sustained for 35 min, prior to the removal of the embolus. After the withdrawal of the embolus, the blood flow returned to the baseline within 10 min (Figure 1a). The 80% reduction in blood flow meant the middle cerebral artery was successfully occluded. Subsequently, we found normal brain tissues could be stained red (TTC staining, Figure 1b) or blue (CV staining, Figure 1c), while the ischemic tissues remained unstained. These results suggest that cerebral ischemia was induced successfully.

Next, we administered Angpt2 or BSA accordingly in the striatum of mice before MCAO or Sham was performed, and then the cerebral ischemic infarct volume changes were explored in the four groups: the BSA and Sham group (BSA-Sham), the Angpt2 and Sham group (Angpt2-Sham), the BSA and MCAO group (BSA-MCAO) and the Angpt2 and MCAO group (Angpt2-MCAO). The results of CV staining are shown in Figure 1d, which shows that neither the BSA-Sham nor the Angpt2-Sham group could cause cerebral infarction, while cerebral infarction is evident in both the BSA-MCAO group and the Angpt2-MCAO group (infarct areas enclosed by dark lines). Then, with image J, we calculated the infarct volume. Representative coronal sections of the infarct area are shown in Figure 1e, and the cerebral infarct volume in the Angpt2-MCAO group was 3.22 ± 0.68 mm^3^. This was significantly lower than that in the BSA-MCAO group of 9.07 ± 1.49 mm^3^, suggesting that Angpt2 could significantly reduce cerebral infarction caused by ischemic injury (Figure 1f). MCAO was performed 12 h after Angpt2 injection into the mice brains. Subsequently, neurological assessments of the BSA-MCAO and Angpt2-MCAO mice were conducted at 24 h and 3 days after MCAO, and the results were as follows: at 24 h, 4/6 of the BSA-MCAO and 2/5 of the Angpt2-MCAO mice showed neurological impairment, with flexion of both the torso and contralateral forelimb when the mouse was lifted by the tail; on day 3, 1/6 of the BSA-MCAO mice showed impairment, but none of the Angpt2-MCAO mice did (Figure 1g). The IHC-immunohistochemistry staining of NeuN^+^ cells in striatum in both the BSA-MCAO and Angpt2-MCAO groups showed, as shown in Figure 1h,i, that the number of NeuN^+^ cells in the BSA-MCAO was 531.20 ± 41.84 mm^2^. This was significantly lower than that in the Angpt2-MCAO of 1042.45 ± 67.68 mm^2^.

### 3.2. Angpt2 Increased CD34 Expression in ECs In Vitro and the CD34^+^ Vascular Length and Area in Mouse Brains after Ischemic Stroke

In order to explore whether the protective function of Angpt2 in stroke is associated with angiogenesis, which is in reference to CD34 expression level, we first evaluated whether ischemic injury could induce CD34 expression in the injured brain regions. The results showed that CD34 expression on the ipsilateral side of ischemia increased significantly (Figure 2a,b) relative to the contralateral ischemia, and immunohistochemistry and immunofluorescence staining further verified that CD34 expression increased in blood vessels after ischemic injury (Figure 2c,d).

Then, we investigated the influence of Angpt2 on CD34 expression on cultured ECs. As shown in the Western blot (Figure 2e,f), 100 ng/mL Angpt2 had significantly increased CD34 expression (1.3 ± 0.09 times) at 30 min, was higher (1.9 ± 0.06 times) at 2 h, and was still high at 6 h (1.75 ± 0.58 times), which indicated that Angpt2 could activate ECs and increase CD34 expression.

Furthermore, to evaluate the effectiveness of Angpt2 in increasing the CD34^+^ blood vessels, we calculated the length and area of the CD34^+^ blood vessels from the three views of the infarction border or needle border by immunofluorescence and compared the BSA-MCAO and Angpt2-MCAO groups. As per the results shown in Figure 2g–i, the length of the CD34^+^ blood vessels (14.1 ± 0.76 mm/mm^2^) in the Angpt2-MCAO group was significantly longer than that of those in the BSA-MCAO group (10.2 ± 0.57 mm/mm^2^); the CD34^+^ blood vessel area in the Angpt2-MCAO group accounted for 2.5 ± 0.18% of the total area of the fluorescence picture, which was significantly larger than the area in the BSA-MCAO group (1.8 ± 0.10%). These results suggest that Angpt2 is closely associated with angiogenesis after stroke.

## 4. Discussion and Conclusions

This study provides the first evidence that Angpt2-triggered a decrease in infarction and neuroprotection, which is closely associated with enhanced CD34 expression and its ability to increase vascular length and area. These findings indicate that Angpt2 promotes angiogenesis and attenuates injury in the brain after stroke, which could be a steppingstone for exploring new potential ways to promote treatments for brain repair through angiogenesis after stroke.

Generally, there are two ways for blood vessels to grow: angiogenesis, forming neovessels from existing vessels; and vasculogenesis, forming vessels from aggregated EPCs. In angiogenesis, a subset of ECs takes on the role of endothelial tip cells, which extend lamellipodia and filopodia into a vascular space [27]. Coincidentally, Angpt2 has a higher expression level in endothelial tip cells. In our research on the effects of Angpt2 in response to ischemic stroke, we showed that the Angpt2-MCAO group showed a significant difference to the BSA-MCAO group in neurological assessment, and Angpt2 played an important role in reducing the ischemic volume and preventing neuronal death, which were confirmed by CV staining and IHC-immunohistochemistry. The function of Angpt2 in ischemic stroke is poorly documented and understood. However, it has been shown that Angpt2 is multifunctional in stroke: on the one hand, it could act as a vasoprotective agent after stroke [6], reducing BBB permeability; on the other, it mediates the differentiation and migration of neural progenitor cells in the subventricular zone after stroke [28]. To date, there have been no reports on Angpt2 in the processes of enhancing angiogenesis after ischemic stroke.

The present study highlights that the exogenous Angpt2 induces angiogenesis through increasing CD34 expression in cultured ECs in vitro and extending CD34^+^ vascular area and length during the acute phase of ischemia. It is well established that CD34 plays an important role in angiogenesis and activated ECs [29]. For example, CD34^+^ cells increase the number of capillary-like tubes formed by human microvascular endothelial cells [30]; while co-culture of HUVEC and CD34^+^ cells increases the sprout length of HUVEC spheroids by 18% [31]. Here, our study found that, after the Angpt2 addition, the expression of CD34 significantly increased in ECs in vitro (Figure 2e,f) and in the brains of mice after stroke (Figure 2a,b). This is consistent with the earlier research that indicated the number of activated ECs increased and participated in the blood vessel repair, subsequently reducing the cerebral damage after stroke with a decrease in infarct volume and neuronal damage. Furthermore, to quantify the angiogenesis induced by Angpt2, we used vascular length and area as the reference. The injection of Angpt2 into the striatum reduced the volume of cerebral infarction (Figure 1d–f) and increased the length and area of CD34^+^ blood vessels per unit area (Figure 2g–i), indicating angiogenesis. Thus, these results further attest to the promotional effects of Angpt2 on angiogenesis after ischemic stroke.

However, some researchers have found that the constitutive overexpression of Angpt2 is associated with brain damage, in contradiction of our results. The reasons for these disparate findings are not clear, but we hypothesize that the function of Angpt2 could be closely related to the types of overexpression: the constitutive overexpression of Angpt2 in transgenic animal models might increase brain damage; however, the administration and adenoviral expression of Angpt2 in cells and animal models, which only function in a short period, might promote protection of the brain after stroke. For example, constitutive overexpressing of Angpt2 in ECs of transgenic mice increased the volume of infarction and enhanced the degradation of pericytes [32]. However, adenoviral expression of Angpt2 induced angiogenesis [33] and administration of Angpt2 before MCAO reduced BBB permeability [6].

In summary, this research demonstrated that Angpt2 contributes significantly to neuroprotection after stroke and enhances angiogenesis, which is associated with CD34^+^ vascular length and area. These results identify a novel role for Angpt2 in brain repair after stroke and could lead to a better understanding of vascular angiogenesis. Our findings also supplement the existing body of knowledge about the role of ECs and angiogenic factors in ischemic stroke, which could be used as a potential strategy for improving stroke treatment and recovery.

## Figures and Tables

**Figure 1 brainsci-12-01428-f001:**
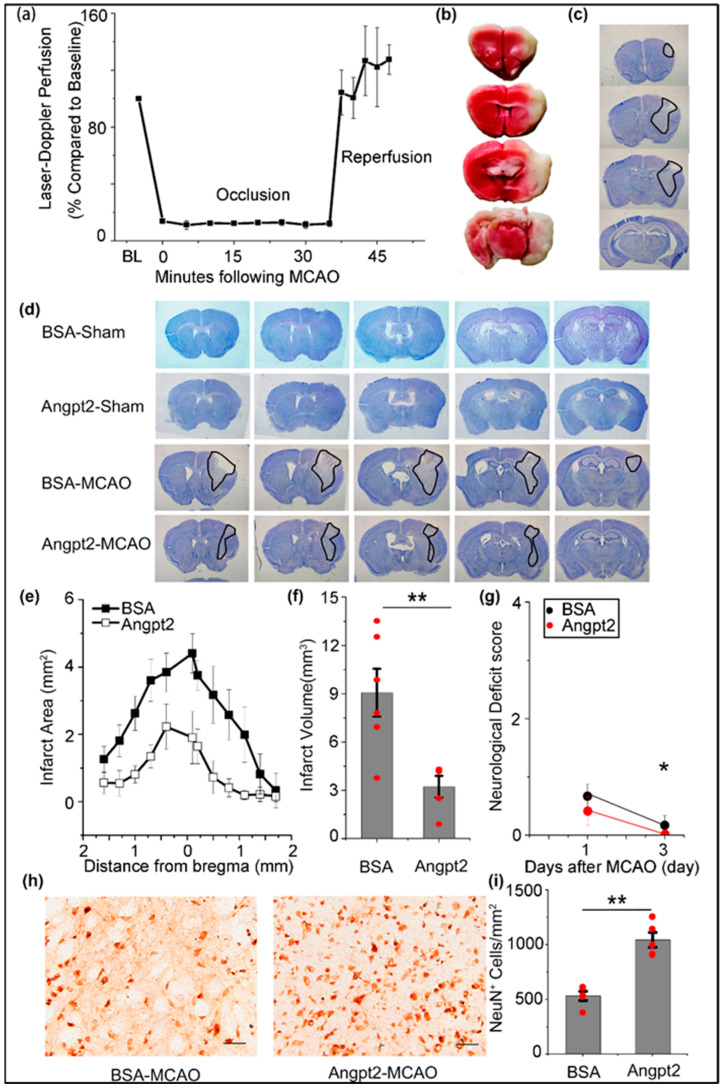
Angiopoietin 2 (Angpt2)exerted a neuroprotective effect by reducing brain infarction in mice brains after ischemic stroke. (**a**) Laser Doppler flowmetry changed through an intraluminal middle cerebral artery occlusion (MCAO) procedure. The baseline blood flow was considered to be 100% for all mice. (**b**) TTC-stained sections of mouse, 3 days after MCAO. The normal brain tissue was stained red, while the ischemic tissue remained unstained. (**c**) Cresyl violet staining in a representative whole section image. The normal brain tissue was stained blue, while the ischemic tissue marked by black lines remained unstained. (**d**) Representative whole section image showing CV staining in the BSA-Sham, Angpt2-Sham, BSA-MCAO and Angpt2-MCAO groups 3 days after surgery. The normal brain tissue was stained blue, while the ischemic tissue remained unstained and marked with black lines. (**e**–**g**) Quantitative analysis was performed to display infarct area distribution (**e**), infarct volume (**f**), and neurological deficits (**g**) between the BSA-MCAO group and Angpt2-MCAO group in mice after stroke. (**h**) Representative images of immunolabeling of NeuN in the brain of mice 3 days after BSA or Angpt2 administration following ischemic stroke. (**i**) The number of NeuN^+^ cells were counted as the average of the number of positive cells in the three fields of view. Lines depict SEM. (*n* = 6 in the BSA-MCAO group; *n* = 5 in the Angpt2-MCAO group; *n* = 5 in h and i; *n* = 4 in other groups; * *p* < 0.05, ** *p* < 0.01 by two-sample *t*-test).

**Figure 2 brainsci-12-01428-f002:**
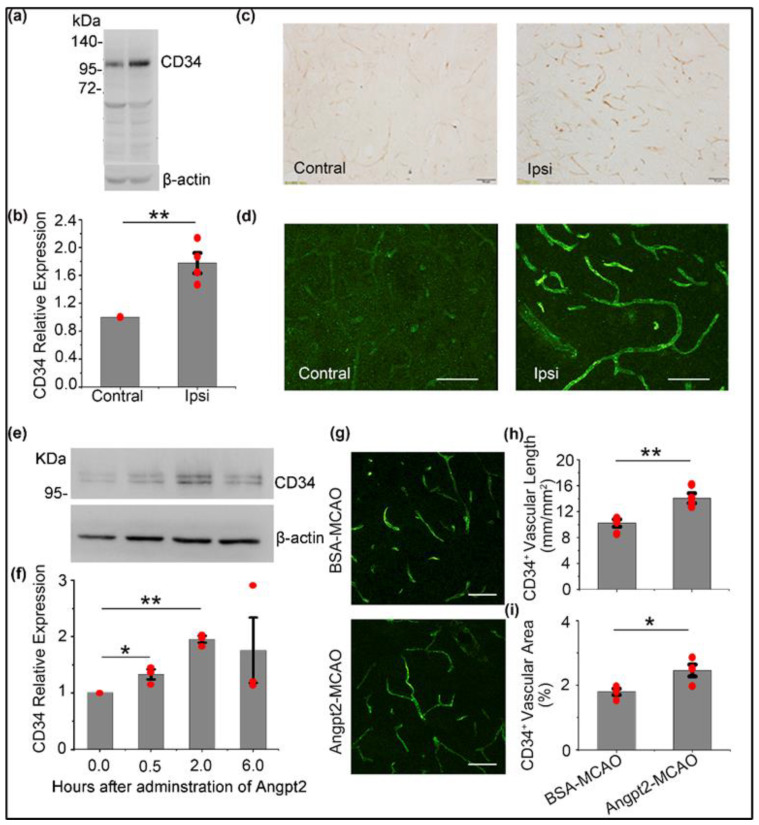
Angpt2 increased CD34 expression in ECs and the CD34^+^ vascular length and area in mouse brain after ischemic stroke. (**a**) Representative immunoblot with CD34 antibody of contralateral (Contral) and ipsilateral (Ipsi) striatum lysates 3 days after MCAO. (**b**) The bar graph described the relative expression of integrated density of the two bands in (**a**). (**c**,**d**) Representative images of immunohistochemistry (**c**) and immunofluorescence (**d**) labeling for CD34 in the ipsilateral and contralateral striatum 3 days after MCAO. (**e**) Representative immunoblot with CD34 antibody of cultured ECs at different time points after Angpt2 addition. (**f**) The bar graph described the relative expression of integrated density of the four bands in (**e**). Lines depict SEM (*n* = 3). (**g**–**i**) Angpt2 increased CD34^+^ vascular length and area after stroke. (**g**) The immunofluorescent image of vessels (green, CD34) in different groups 3 days after injection and operation. The statistics of CD34^+^ vascular length (**h**) and area (**i**) in different groups 3 days after injection and operation. Scale bars: 50 μm. Lines depict SEM. (*n* = 3 in e and f; *n* = 4 in other groups; In b, h and i, * *p* < 0.05, ** *p* < 0.01 by two-sample *t* test. In f, * *p* < 0.05, ** *p* < 0.01 by one-way AVONA with Tukey’s post hoc test).

## Data Availability

Not applicable.

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
