# Peer review of "Neuroprotective Effect of Angiopoietin2 Is Associated with Angiogenesis in Mouse Brain Following Ischemic Stroke"

_brainsci, 2022, doi:10.3390/brainsci12111428_

Round 1
Reviewer 1 Report
The study Neuroprotective Effect of Angiopoietin2 is Associated with Angiogenesis in Mouse Brain Following Ischemic Stroke is really interesting.
You were able to provide really interesting results, which are helpful to understand what we can do to mitigate the impact of stroke.
It would be interesting to go further in your analysis.
Minor:
Introduction: It would be interesting to clearly state the hypothesis in the intro section.
Materials and methods: What is the age of animals and the ethic protocol #
Major
Results:
It would be interesting to show if the treatment protects the BBB integrity since you show an amelioration in the number of blood vessels.
You can show it by WB.
It would also be interesting to qualitatively or quantitatively analyze the microglia around the vessels and the secretions of those cells which are really important for the BBB integrity and the blood vessels as well.
I check "reconsider after major revision" not because the study lacks of control or experiments but to encourage you to dig deeper and provide more evidence.
Reviewer 2 Report
I have received the research paper entitled “Neuroprotective Effect of Angiopoietin2 is Associated with Angiogenesis in Mouse Brain Following Ischemic Stroke” by Ling-Ling Lv et al. for evaluation. In this manuscript, the authors have shown that Angiopoietin2 (a secreted glycoprotein) improves the angiogenesis in focal cerebral ischemia in mice. The study is timely and important. However, the authors have performed fewer experiments over the extrapolated result. My critics and comments are given below.
Measure comments:
1- The authors have concluded that Angiopoietin2 is neuroprotective against cerebral ischemia without generating histopathological data showing neurodegeneration. I would suggest the author confirm the Angiopoietin2 -mediated neuroprotective against cerebral ischemia by adapting the immuno-staining technique [neuronal marker; NeuN staining (Singh et al. 2012; PMID: 20371137)].
2- Figure 1 (g) showing the neurological deficit score is not very clear. What is the assay name, procedure of assay and Which type of instrument was used? What is the unit of assay score?
3- I would suggest authors plot all the data set in the bar diagram instead of showing only the Mean value (please reference Tiwari et al., 2022; PMID: 35732494)
Minor Comment:
1- The methodology section required extensive revision. Many sections are without citations.
2- The Neurological Assessment section should be more elaborative and cited with appropriate references.
3- In the Materials and Methods section, the statistical analysis section is missing.
4- The figure legend of each figure must have the name of statistical
Round 2
Reviewer 1 Report
Thank you for the modifications
Reviewer 2 Report
Authors have addressed all the comments raised by me. I recommend that manuscript may be accepted in present revised format.